# Adipose Autologous Micrograft and Its Derived Mesenchymal Stem Cells in a Bio Cross-Linked Hyaluronic Acid Scaffold for Correction Deep Wrinkles, Facial Depressions, Scars, Face Dermis and Its Regenerations: A Pilot Study and Cases Report

**DOI:** 10.3390/medicina58111692

**Published:** 2022-11-21

**Authors:** Lorenzo Svolacchia, Claudia Prisco, Federica Giuzio, Fabiano Svolacchia

**Affiliations:** 1Departments of Medical-Surgical Sciences and Biotechnologies, Sapienza University of Rome, 00118 Rome, Italy; 2Department of Experimental Medicine, Università degli Studi della Campania Luigi Vanvitelli, 80138 Naples, Italy; 3Department of Science, University of Basilicata, 85100 Potenza, Italy; 4Departments of Medical Sciences, Policlinico Tor Vergata Foundation, Tor Vergata University of Rome, 00133 Rome, Italy

**Keywords:** adipose autologous micrograft, cluster of differentiation (CD), dermal regeneration, hyaluronic acid, side population, tissue progenitor

## Abstract

The aim of this clinical study was to demonstrate that through a micrograft of viable adipose tissue cells microfiltered at 50 microns to exclude fibrous shoots and cell debris in a suspension of cross-linked hyaluronic acid, we were able to improve visible imperfections of the dermis and to improve clinically observable wrinkles, with a beneficial effect also in the extracellular matrix (ECM). *Background and Objectives*: With the passage of time, the aging process begins, resulting in a progressive impairment of tissue homeostasis. The main reason for the formation of wrinkles is the involution of the papillary dermis, as well as the loss of stem cell niches with compromise of the extra-cytoplasmic matrix (ECM), and the loss of hyaluronic acid, which helps to maintain the shape and resistance and that is contained in the connective tissue. *Materials and Methods*: This study involved 14 female patients who underwent dermal wrinkle correction and bio-regeneration over the entire facial area through a suspension containing 1.0 mL of viable micrografts from adipose tissue in a 1.0 mL cross-linked hyaluronic acid. To verify the improvement of the anatomical area concerned over time, the various degrees of correction obtained for wrinkles, and in general for texture, were objectively evaluated by using a Numeric Rating scale (NRS) 10–0, a modified Vancouver scale and a Berardesca scale. *Results*: The Berardesca, NRS and Modified Vancouver scales showed that with this technique it was possible to obtain excellent results both when the suspension was injected into wrinkles with the linear retrograde technique, and when it was injected with the micropomphs technique to correct furrows, with the intent to revitalize the tissue through progenitors with adult stemness markers. *Conclusions*: The combination of microfragmented and microfiltered adipose tissue and cross-linked hyaluronic acid at 50 microns is safe new method to treat soft tissue defects such as deep wrinkles.

## 1. Introduction

Correct tissue homeostasis and balance are essential to maintaining the physiological cell turnover in tissues and in the extracellular matrix (ECM). Exogenous and endogenous agents interfere with homeostatic efficiency and decrease the ability of cells in more complex tissues to maintain their integrity by altering structural molecules that are necessary for them [1]. Morphological characteristics of senescent cells are the reduction in cell volume and the accumulation of lipofuscins that are responsible for the hypotrophy of tissues in conjunction with the slowdown in cell multiplication [2]. Therefore, the generation of free radicals inside cells and the accumulation of defective molecules lead to an impairment of their biological function [3]. Mutations due to free radicals also affect mitochondrial DNA, which leads to a malfunction of those proteins involved in the production of ATP with the consequent alteration of the bioenergetics of involved cells [3]. Modifications induced by the compromise of anchoring systems of cells that are ubiquitous in all tissues are also involved in the aging process [4]. The consequence of these pathophysiological processes allows an increase in tissues of proinflammatory cytokines produced by dendritic cells and macrophage tissue, with the activation of the pro-inflammatory genetic program NF-Kb, which is also responsible for maintaining an inflammatory state over time [4]. Therefore, the result will be an induction of replicative senescence, activation of aginggenes, and the loss of niches of Adult Stem Cells, with an involution of tissue homeostasis [5].

The phenomenon of dermal aging is also closely related to pathologies in tissue elastin, which can only be acquired and triggered by the inflammation that leads to the deterioration of elastic fibers [4]. This is generally associated with tissue fibrosis, which constitutes an increased deposition in the tissues of collagen produced by cells in a non-physiological way. Both of these pathophysiological events contribute to accentuate the pathophysiological phenomenon of aging of the dermis [5]. Cytokines most commonly involved in the process of genesis of a fibrotic tissue are TGF-β, IL-1, TNF-α, and PDGF, while the most important cell in the formation of fibrosis is the fibroblast, which, through the inflammatory stimulus, encodes the formation of type I collagen and the consequent modification of the extracellular matrix by compromising the anchoring systems of cells within the ECM [6]. 

Through an increase in fibronectin and laminin, the bonds with type I collagen are stabilized, and through the integrin system they help to generate new signals of fibrosis induction by means of an accumulation of proteins that are inappropriate for that tissue and capable of stimulating specific alarms such as HGMB-1. The consequence will be an increase of inflammatory signals towards the NF-Kb genetic program with an increase in the impairment of the ECM [7]. Its degradation will consequently be modulated through the activity of specific glycoproteins, the Metalloproteinases (MMPs), which after being secreted provide an important contribution to fibrotic tissue remodeling [8]. MMPs can be inhibited by metalloproteinase inhibitors (TIMPs). Adult stem cells that are contained in adult tissues in a variable percentage from 0.001 to 1%, protected by specific niches and randomly arranged, can locally intervene in the cell turnover process that characterizes the physiological homeostasis of tissues. They participate in the regenerative processes and secrete inhibitors of metalloproteinases: TIMPs, and inhibitors of plasminogen activators: PAISc [9]. The latter is produced by adult mesenchymal cells that participate in regenerative processes, restore integrity by maintaining the tissue architecture by preventing an alteration of the geometry of tissues in the papillary dermis, and by also preventing an increase in the distance between cells and vessels by avoiding the consequent diffusional hypoxia [9]. The replacement of native cells to maintain a physiological homeostasis and the integrity of the tissues appears to be conditioned by the amount of intact adult stem cells, and by the efficiency of telomerase, while somatic mutations of adult stem cells and the depletion of niches are closely related to organ-tissue aging [10]. In this context, it is fundamental the supplementation of new cells that have the characteristics of adult stem cells, in order to locally intervene in the cell turnover processes, which characterizes a physiological regenerative process [11].

Hyaluronic acid is one of the main components of connective tissues, in humans it helps to maintain their shape and resistance, but its progressive depletion in tissues it contributes to the loss of volumes and to the formation of wrinkles. The natural and most important ligand for hyaluronic acid is CD44, a highly glycosylated transmembrane protein composed of 177 amino acids, and that it is able to induce survival mechanisms in certain cell, through an obligatory bond called binding [12,13]. CD44 is commonly expressed on the surface of adipose-derived adult mesenchymal stem cells [14,15] and it can also bind fibronectin and selectin by improving homeostatic regulation and physiological responses to growth factors [16].

The union of the cross-linked hyaluronic acid with stem cells or progenitors for a regenerative therapy of the dermis would allow a physiological neo-collagen genesis through the activation of CD 44 [14]. This can be boosted by the mechanical properties of hyaluronic acid with a synergistic action of filling and regeneration of the treated tissue [17]. This technique should increase the vitality and longevity of the progenitors injected with the linear retrograde technique or micro wheal, and it is linked to the re-viable cells and their consequent improved functionality [18].

An increase in both qualitative and quantitative biological parameters would have allowed an improvement in the plastic characteristics of progenitors through an increase in the number of new cells, in conjunction with a long-lasting scaffold caused by cross-linked hyaluronic acid. These phenomena would have consequently allowed an increase in cell turnover in the dermis, which would have induced a clinically-observable increase in the secretion of physiologically expressed collagen and elastin [19].

Initially, this hypothesis had been already demonstrated in another study with non-crosslinked hyaluronic acid [20]. which opened a new perspective on the association between cross-linked hyaluronic acid and a small amount of adipose tissue. The emulsification according to Tonnard 2013 and microfiltering, allow the deprivation of the inflammatory component derived from fibrous shoots and cellular debris [21] that are potentially able to activate the Toll-Like system 4 [22]. We have shown that filtration of disintegrated adipose tissue at certain sizes allows for the maintenance of a viable and-numerically speaking, a very high side population, even within 2 mL of processed tissue [23]. It is certain during the disintegration, according to Tonnard 2013, but especially during filtration, that there is a loss of vital elements [24], but their therapeutic potential is higher [25] because fibrous shoots and cellular debris are eliminated.

The exclusion of fibrous shoots and cell debris prevents ADSCa from expressing inflammatory Toll-Like Receptors, and thus from recruiting additional inflammatory cells to the implantation site [26]. In fact, the activation of TLR4 on them it would lead to the selection of proinflammatory adult mesenchymal stem cells, with proinflammatory secretomes [21] with a subsequent modulation of the inflammation and an activation of the fibroblastic process [27] that are incompatible with a system that provides tissue regeneration and with our studio. Therefore, the clinical hypothesis was to correct the soft tissue damage through a combination of a durable scaffold like hyaluronic acid mixed with progenitors derived from adipose tissue [28]. This strategy might promote the survival of the progenitors, and to allow them to resist the aggression of exogenous and endogenous agents [29]. 

## 2. Patients and Materials

The study was approved by the local Ethics Committee with authorization No. 11032/2022. A total of 14 patients (aged between 41 and 58 years) were included in the study. Inclusion criteria were the following:Male or female subjects between 30 years and 65 years;The presence of deep wrinkles in the facial area, e.g., around the eyes, around the mouth and cheeks, including wrinkles and spots on the face;Skin free of diseases that could interfere with the evaluation of the results;Willingness to abstain from any cosmetic or surgical procedures in the treatment area for theduration of the clinical investigation;Willingness to abstain from any facial surgical procedures for the duration of the clinical;investigation including application of botulinum toxin;Willingness to abstain from excessive weight gain or weight loss (±10% of body weight), and/or drastic dietary changes for the duration of the clinical investigation;Written informed consent.

### 2.1. Exclusion Criteria

For females: pregnancy, lactating, planned pregnancy;History of mental disorders or emotional instability;History of allergic reaction to HA products;Facial surgery or implantation of dermal fillers in the nasolabial region within the last 24 months;Skin of the to be treated region affected by cosmetic treatments (e.g., laser therapy within the last 12 months, chemical peeling within the last 3 months, and dermabrasion within the last 12 months, and botulinum toxin within the last 12 months);Connective tissue diseases;Diabetes mellitus or uncontrolled systemic diseases;Known human immune deficiency virus-positive individuals;Presence of silicone implants or implants of another non-absorbable substance (permanent fillers) in the area of product application;Cutaneous lesions in the evaluated area;Tendency to keloid formation and/or hypertrophic scars;Autoimmune disease;Any medical condition prohibiting inclusion in the study according to the judgment of the investigator;Subjects for whom due to a mental disorder or a mental disability a custodian has been appointed or who are legally or magisterially arrested or housed;Current or previous (within 30 days of enrollment) treatment with an investigational drug and/or medical device or participation in another clinical study;History of allergies to cosmetic filling products and recurrent herpes simplex virus;Heavy smokers (>20 cigarettes per day).

### 2.2. Procedure

The study was performed by following the standards of the local ethics committee, and in accordance with the Declaration of Helsinki (2000). Patients were subjected to ta subdermal injection of a suspension containing 1 mL of viable micrografts derived from adipose tissue [30] and emulsified in 1 mL of cross-linked hyaluronic acid as a scaffold. Adipose tissue was extracted following the protocol provided by the MilliGraft kit^®^ (Dual Trend srl Corso Torino, Chieri, Italy). This is a standardized protocol that guarantees the correct extraction of numerous progenitors [30,31]. In fact, the disposable kit contains everything necessary for a quick and simple extraction and processing of the necessary quantity of adipose tissue, in addition to 50-micron filters. With these materials we have been able to obtain vital progenitors in the suspension by excluding fibrous shoots and potentially-inflammatory cellular debris [30]. The procedure to obtain a fat micrograft with the MilliGraft kit^®^ includes four steps, as follows: (1) A regional site anesthesia is performed. The fat tissue is then removed. Both procedures are carried out through the use of a needle connected to a Luerlock^®^ syringe to simplify the procedure; (2) the removed tissue is immediately processed in order to get a gradual reduction of adipose clusters, and a MilliGraft kit is used to obtain a fluid suspension without oily and hematic components that are usually pro-inflammatory; (3) The processed adipose tissue is filtered with 50μm mesh in order to isolate a cell population without debris [32]. Such obtained solution (Figure 1B) was mixed and cross-linked HA with Panthenol “Dermal plus 25 High Performance” 1 mL prefilled syringe from Beauty system Pharma Srl, (lot 0219820) to serve as a scaffold (Figure 1A). The sterility of the final product was maintained as the MilliGraft kit^®^ guarantees a sterile final solution thanks to a closed system which was mixed with sterile hyaluronic acid by connecting the two syringes (Figure 1C,D). All of these procedures were carried on in a surgical theater.

### 2.3. Detailed Procedure

After administering local anesthesia to the abdomen or supratrocanteric donor area with Klein’s solution through a 10 mL syringe and a 25 G needle, the extraction of a total of 3 mL of lipoaspirate was undertaken. 

The extraction with a needle allows a better survival of progenitors [30], as well as an overlap in the quantity and quality of viable extracted cells, rather than the extraction with a multi-hole cannula [24,25]. The use of a needle makes the procedure very quick. After the extraction of the adipose tissue, the suspension was left to settle for 10 min, in order to eliminate anesthesia fluids. Two mL of adipose tissue was then processed as described by Tonnard 2013 [31], and filtered at 50 microns to preserve the cell population [30], from which about 1.2 mL of a final suspension was obtained (Figure 1B). It is known that during the disintegration and the subsequent filtration, a loss of vital elements may occur [25]. Nevertheless, by eliminating fibrous shoots and cellular debris, the therapeutic quality of progenitors is improved [26], as fibrous and cell debris could activate an inflammatory process through the Toll-Like system [30]. The microfiltrate was divided into two 1 mL syringes and added to 0.5 mL of hyaluronic acid through a three-way tap with a very gentle back and forth movement (made for 5–7 times) in order to emulsify the two parts (Figure 1C,D). The obtained suspension remained stable during the injection treatment, without any settling of parts occurring during the procedure. The suspension was administered through a 25 G needle for the deep dermis, and with a 30 G needle for more superficial treatments. Techniques that we used were those of retrograde injection and micropomphs. The entire procedure took 45 min for each patient. 

### 2.4. Follow-Ups

Two different scoring systems were used to assess the efficacy of the studied procedure, the patients’ satisfactory score proposed by Berardesca et al. [33] and the Vancouver Scar Scale [34] that we used to estimate the general improvement of the skin appearance, considering four parameters: ductility, height, vascularity, and pigmentation. The patients were subject to examination and scoring at day 0 (baseline), day 1 (follow up 1), day 80 (follow up 2), and day 150 (follow up 3). The data were collected and analyzed at the end of the study.

## 3. Results

All patients were female and did not have any specific dermal pathologies or other systemic pathologies that were not-pharmacologically controlled. A progressive and significant improvement in skin relief can be observed during treatment. In comparison to D0 (before treatment), after 1 day, 80 and 150 days, subjects satisfaction was evaluated by giving scores on firmness and cutaneous relief by using a scale of 0–4 for each criterion (0 = unsatisfactory; 4 = satisfactory), as suggested by Berardesca et al. [33]. As expected, the data shown in Figure 2 demonstrate a high degree of satisfaction that lasts over time (5 months). In addition, individual wrinkle signs, as well as the defect severity obtained for each treatment and each area, were objectively evaluated by using a 10–0 Numeric Rating Scale (NRS) with separate scores for each site (10 = High wrinkle signs or High defect severity; 5 = Medium wrinkle signs or medium defect severity; 0 = Low wrinkle signs or medium defect severity). This scale is our creation and aims to give a numerical measure of the degree of severity of the general facial defect and more specifically of wrinkle severity, before starting the treatment (D0), during the treatment (D1 or D80), and at the end of treatment (D150). Results shown in Figure 3 demonstrate that the treatment with viable micrografts derived from adipose tissue and cross-linked hyaluronic acid induces a significant reduction of the wrinkle signs and defects severity, in all patients. Tissues treated with both the retrograde and micropomph techniques were also evaluated through the modified Vancouver scale: parameters analyzed are ductility, height, vascularity and pigmentation, as shown in Figure 4. The Vancouver Scar Scale is widely used in clinical practice and research to document changes in scar appearance, and in our study we used that to estimate in general the improvement of the skin appearance considering the four parameters mentioned above (ductility, height, vascularity and pigmentation). The results from the Vancouver scale show that all the paraments considered significantly decreased at day 80 and 150. These results are consistent with the improvement of the treated sites. A representative image of the results at day 80 are represented in Figure 5. Moreover, treatment safety was recorded and the use of the product did not lead to any unwanted cutaneous reaction, demonstrating its complete safety. 

## 4. Discussion

Human mesenchymal stem cells (hMSCs) have been presented as a promising cell source for regenerative medicine in a variety of settings, including bone and cartilage repair, cardiac, vascular, neuronal, and endocrine rescue. These cells can self-renew at a rapid pace and have multipotent differentiation abilities. According to in vitro, ex vivo, and in vivo data, these multipotent cells may develop into mature adipocytes as well as chondrocytes, osteoblasts, myocytes, hepatocytes, neuronal-like and endothelial cells, and other lineages, and this potential may be employed to heal damaged tissues.

Furthermore, MSCs release a number of bioactive chemicals that function in a paracrine fashion to trigger and maintain angiogenic, antifibrotic, antiapoptotic, and immunomodulatory responses in target tissue [35]. Adipose tissue is an excellent source of MSCs, as the number of these cells in this tissue is very high compared to other tissues, such as the often-used bone marrow. Moreover, adipose tissue can be easily obtained with a very low impact on patients’ health. Adipose derived stem cells (ASCs), like MSCs generated from bone marrow, are multipotent and capable of differentiating into mesenchymal lineages. Nonetheless, they have similar immunophenotypes, differentiation abilities, proteomes, immunomodulatory capabilities, and transcriptomes [11]. The last years have seen the development of many non-enzymatic methods to obtain a solution containing ASCs. These methods avoid the problems and costs GMP facilities and can be performed in the surgery theater without special equipment. The MilliGraft kit^®^ belongs to this family of tools and has been used in several applications [36]. Microfragmented adipose tissue such as the one obtained with MilliGraft has been shown to contain good quality MSCs [37]. Microfragmented adipose tissue has been shown to have anti-inflammatory [38] and regenerative properties and it has been used to treat diseases such as osteoarthritis [39,40], Crohn’s disease fistula [41], and others. Adipose derived products such as nanofat and sìthe and so called stromal vascular fraction SVF has been also applied in aesthetic and reconstructive medicine, mostly for scar reduction and facial rejuvenation [42] Nevertheless, the combination of filtered microfragmented adipose tissue with cross-linked hyaluronic acid has never been used to treat soft tissue defects. Thus, in this work we challenged this hypothesis. Our aim was to clinically demonstrate that this combination could be used to improve the regenerative process of soft tissue for the correction of deep wrinkles and to obtain a general improvement of the treated sites. Our study involved 14 female volunteer patients (aged between 41 and 58 years), and the only criterion for entry into the study was the presence of deep wrinkles. A progressive and significant improvement in skin relief can be observed during treatment. In comparison to D0 (before treatment), after 1 day, 80 and 150 days, subjects evaluated their satisfaction by giving scores on firmness and cutaneous relief by using a scale of 0–4 for each criterion (0 = unsatisfactory; 4 = satisfactory). All patients were satisfied with the treatment. The physical examination that patients underwent during the follow-up was in line with the self-assessment. This study must be considered as being very preliminary, with the aim to show that the technique proposed is safe and the results consistent. Additional studies with more patients and a randomized structure are needed to assess the benefit of this new method. In vitro and in silico study might also be performed to clarify the mechanism of the interaction between biomaterials and cells contained in the microfragmented adipose tissue. Nevertheless, this research provides a new opportunity to study the combination between microfragmented adipose tissue and biomaterials in the treatment of soft tissue defects [43].

## 5. Conclusions

This retrospective clinical evaluation study for the treatment of blemishes and dermal regeneration through a suspension of progenitors with adult stem characteristics (ASC) in a scaffold of cross-linked hyaluronic acid allowed us to evaluate the excellent results obtained by the means of this method. This allows us to obtain beneficial effects on the regulation of ECM, neocollagenogenesis, neoelastogenesis and neovasculogenesis. The illustrated method opens a new therapeutic front, not only for the dermis, because it combines the volumetric correction of crosslinked hyaluronic acid with the regenerative one of vital adipose micrografts, to which the inflammatory component constituted by fibrous shoots and cellular debris has been excluded. The processes are quick and provide extreme satisfaction for both the doctor and the patient. We are sure that this new medical practice can also be used for other pathologies, such as the osteo-articular. However, further studies are necessary.

## Figures and Tables

**Figure 1 medicina-58-01692-f001:**
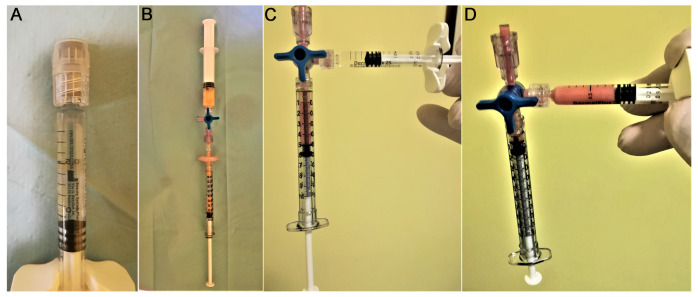
Procedure to mix microfragmented fat with hyaluronic acid. (**A**) Hyaluronic acid prefilled syringe; (**B**) Filtration of the collected and fragmented fat tissue; (**C**,**D**) mixing of the fragmented adipose tissue with hyaluronic acid in a closed system to guarantee sterility.

**Figure 2 medicina-58-01692-f002:**
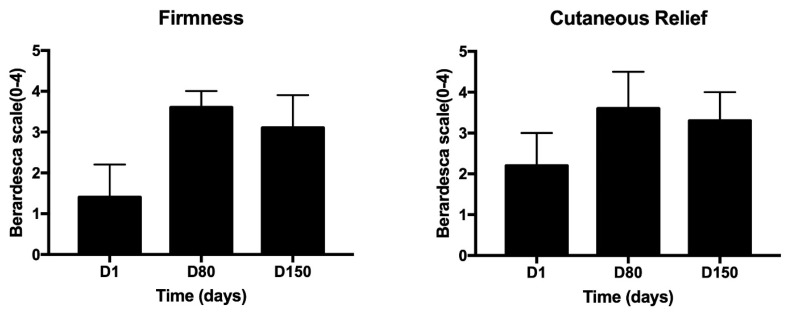
Berardesca Scale for the patient’s satisfaction evaluation. Subjects’ evaluation of their satisfaction in comparison to D0 (before treatment), after 1 day, 80 and 150 days, by giving scores on firmness and cutaneous relief. Scale of 0–4 for each criterion (0 = unsatisfactory; 4 satisfactory).

**Figure 3 medicina-58-01692-f003:**
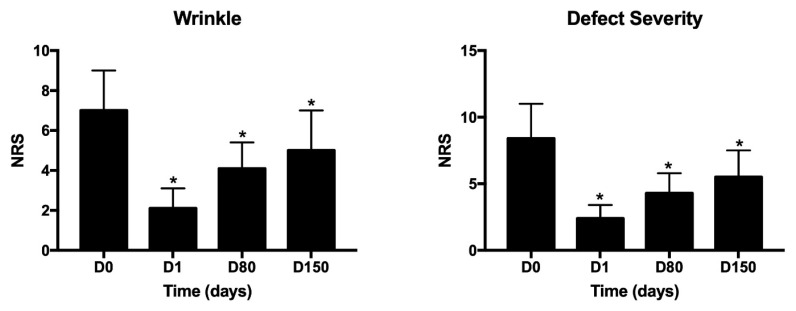
Numeric Rating Scale (NRS) evaluating defect severity and wrinkles. 10–0 Numeric Rating Scale (NRS) with separate scores for each site (10 = High wrinkle signs or High defect severity; 5 = Medium wrinkle signs or medium defect severity; 0 = Low wrinkle signs or medium defect severity); * *p* < 0.05.

**Figure 4 medicina-58-01692-f004:**
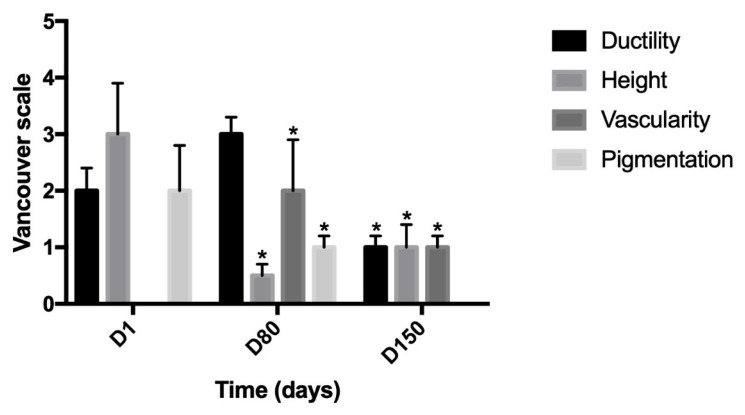
Modified Vancouver scale for the evaluation of ductility, height, vascularity and pigmentation. Modified Vancouver Scale used to estimate the improvement of the skin appearance in comparison to D0 (before treatment), after 1 day, 80 and 150 days. The parameters considered are ductility, height, vascularity and pigmentation. * *p* > 0.05.

**Figure 5 medicina-58-01692-f005:**
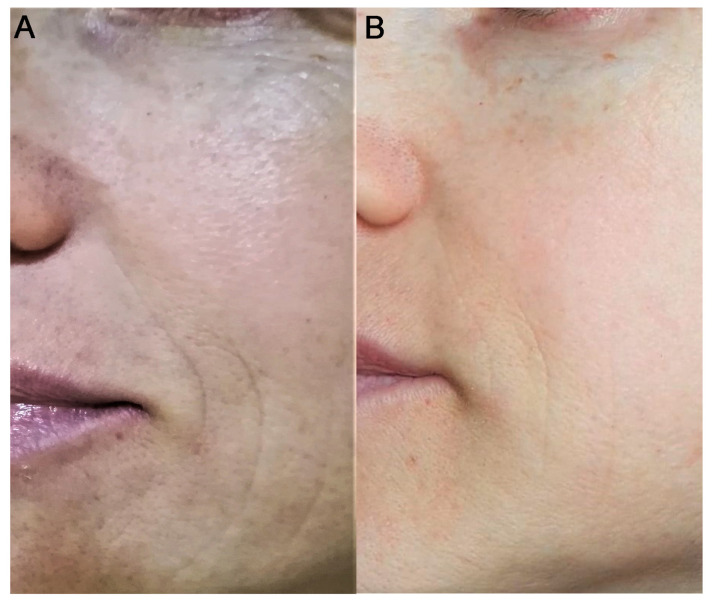
Example of treatment on a patient. (**A**) before treatment; (**B**) 80 days after treatment.

## Data Availability

Not applicable.

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
