# Peer review of "Adipose Autologous Micrograft and Its Derived Mesenchymal Stem Cells in a Bio Cross-Linked Hyaluronic Acid Scaffold for Correction Deep Wrinkles, Facial Depressions, Scars, Face Dermis and Its Regenerations: A Pilot Study and Cases Report"

_medicina, 2022, doi:10.3390/medicina58111692_

Round 1

Reviewer 1 Report

1.      Title is too long and hard to understand, please make it shorter.

2.      In the abstract section, quantitative findings should be reported.

3.      Please end your abstract with a "take-home" message.

4.      Put the keywords in a new order based on alphabetical order.

5.      It is encouraged not used abbreviations in the keywords section.

6.      In the present form, nothing really novel. The current study appears to be a replication or modified study according to the lack of novelty. The authors must extensively describe the novel their work is. This work should be rejected due to a serious concern.

7.      The work, novelty, and limitations of similar prior studies must be explained in the introduction section to highlight the research gaps that the current study aims to fill.

8.      In the last paragraph of the introduction, please specifically explain the objective of the present article.

9.      The authors need to justify why present study perform clinical (in vivo). Also, compaction with experimental (in vitro) and computational (in silico) is recommended. The future study adopting in silico study needs to be mention. in It is a vital topic that authors must provide in the introduction and/or discussion section. Additionally, the MDPI's suggested reverence should be taken to substantiate this explanation as follows: Jamari, J.; Ammarullah, M. I.; Santoso, G.; Sugiharto, S.; Supriyono, T.; van der Heide, E. In Silico Contact Pressure of Metal-on-Metal Total Hip Implant with Different Materials Subjected to Gait Loading. Metals (Basel). 2022, 12, 1241. https://doi.org/10.3390/met12081241

10.   To help the reader grasp the study's workflow more easily, the authors could include more visuals to the materials and methods section in the form of figures rather than sticking with the text that now predominates.

11.   What is the baseline of patient selection? Is there any protocol, standard, or basis that has been followed? It is unclear since the patient is very heterogeneous with a small number. The resonance involved impacts the present result makes this study flaws. One major reason for rejecting this paper.

12.   The error and tolerance of the experimental tools employed in this investigation are critical details that must be explained in the publication. It would be a valuable discussion because of the differing outcomes in the subsequent study by other researchers.

13.   An evaluation of the findings with similar past research is essential.

14.   Discussion is very poor, other from elaborate my comments number 9, another comprehensive discussion is mandatory. Not just simply explain the results.

15.   The limitation of the current study must be included at the end of the discussion section.

16.   In the conclusion, please explain the further research.

17.   The authors should give additional references from the five-years back. MDPI reference strongly recommended.

18.   The authors were encouraged to proofread their work due to grammatical problems and linguistic style.

19.   The authors need to provide a graphical abstract for submission after the revision stage.

Author Response

Response to reviewer 1 :

Reviewer 2 Report

The Reviewer would like to thank the authors for their efforts in performing the study. Attached below are the reviewer's comments.

Author Response

Response to reviewer 2 

Round 2

Reviewer 1 Report

Nice work from the authors.

Reviewer 2 Report

The reviewer would like to thank the authors for their efforts in modifying the manuscript.